# The Impact of a Structured Outpatient Parenteral Antimicrobial Therapy (OPAT) Programme on Quality of Care, Optimisation of Antimicrobial Use, and Healthcare Costs: A Retrospective Cohort Study

**DOI:** 10.3390/antibiotics14111103

**Published:** 2025-11-02

**Authors:** Irene G. Manders, Darya Comello, Dennis Souverein, Sjoerd Euser, Bjorn L. Herpers, Judith Vetten, Jayant S. Kalpoe, Marco Goeijenbier, Steven F. L. van Lelyveld

**Affiliations:** 1Regional Public Health Laboratory Kennemerland, Boerhaavelaan 26, 2035 RC Haarlem, The Netherlands; daryacomello@gmail.com (D.C.); d.souverein@streeklabhaarlem.nl (D.S.); s.euser@streeklabhaarlem.nl (S.E.); j.kalpoe@streeklabhaarlem.nl (J.S.K.); 2Outpatient Pharmacy, Spaarne Gasthuis, Boerhaavelaan 22, 2035 RC Haarlem, The Netherlands; jvetten@sahz.nl; 3Department of Intensive Care Medicine, Spaarne Gasthuis, Boerhaavelaan 22, 2035 RC Haarlem, The Netherlands; mgoeijenbier@spaarnegasthuis.nl; 4Department of Internal Medicine, Spaarne Gasthuis, Boerhaavelaan 22, 2035 RC Haarlem, The Netherlands; s.van.lelyveld@spaarnegasthuis.nl

**Keywords:** outpatient parenteral antimicrobial therapy (OPAT), costs, infectious disease specialist, ID specialist, antimicrobial stewardship, ambulatory care

## Abstract

**Background/Objectives:** In 2022, Spaarne Gasthuis hospital implemented an outpatient parenteral antimicrobial therapy (OPAT) programme, including mandatory infectious disease (ID) specialist assessment and integrated structured workflow, aiming to improve quality of care, optimize antimicrobial use, and reduce healthcare costs. Our objective was to evaluate the impact of the OPAT programme on patient outcomes (IV duration, clinical response, adverse clinical outcomes, timely peripherally inserted central catheter (PICC) removal), antimicrobial stewardship parameters, and healthcare costs. **Methods:** This retrospective before–after cohort study used electronic health record data to compare patients treated with outpatient parenteral antimicrobial therapy before (2019) and after (August 2022–December 2024) OPAT programme implementation. Using linear and logistic regression analyses, the association between the independent variable (pre-OPAT vs. OPAT) and outcomes were assessed and adjusted for potential confounders (sex, age, department, primary and additional indications). Cost analysis was performed, and ID specialist-recommended therapy adjustments were evaluated. **Results:** In total, 529 patients were included: 118 in the pre-OPAT group, and 411 in the OPAT group. In 36.3% of OPAT cases, therapy was adjusted, thereby optimizing antimicrobial stewardship. The OPAT programme was associated with significantly shorter IV therapy duration of 13.97 (mean) days (95%CI −9.15–−18.79; *p* < 0.001), significantly less meropenem use (*p* < 0.001), and significantly less adverse clinical outcomes (OR 0.58, 95%CI 0.37–0.92; *p* = 0.021), whereas no significant difference was found in clinical response (OR 1.22; 95%CI 0.67–2.32; *p* = 0.527). The programme led to cost savings of 3.343 EUR per patient. **Conclusions:** The OPAT programme optimized antimicrobial use and reduced IV therapy duration, adverse clinical outcomes, and healthcare costs.

## 1. Introduction

Outpatient parenteral antimicrobial therapy (OPAT) allows patients to receive intravenous (IV) or intraperitoneal antimicrobial treatment outside hospital settings, aiming to deliver efficient, cost-effective care, while maintaining clinical effectiveness and improving quality of life [1,2,3,4]. OPAT can be delivered in various outpatient settings, including patients’ homes, rehabilitation centres, and long-term care facilities [4,5]. OPAT is commonly used to treat infections that require prolonged IV antimicrobial therapy, including bone and joint infections, endocarditis, device-related infections, and neurological infections [4,6]. It is also used to treat various infections with multidrug-resistant organisms (MDROs) [4,6].

Multiple studies confirm OPAT’s safety, effectiveness, and cost benefits [1,2,3,4,7,8,9,10]. In the United Kingdom, OPAT costs were 41% of comparable inpatient care, with high treatment success, low complication rates, and a low readmission rate of only 6.3% [2]. Nevertheless, structured implementation is limited [11]; only a minority of hospitals in the world have dedicated OPAT teams (in 42%) [12], selection criteria (in 56.4%), and follow-up protocols (61.8%) [13]. Among hospitals with OPAT teams, 78% have an infectious disease (ID) specialist involved [12].

In August 2022, the Spaarne Gasthuis (SG), a large regional teaching hospital in The Netherlands, launched a structured OPAT programme in collaboration with the Regional Public Health Laboratory Kennemerland (RPHLK). Prior to this, outpatient antimicrobial care was provided, yet lacked systematic approach and mandatory assessment by an ID specialist or medical microbiologist. Several limitations of the previously used unstructured approach were noticed: (1) lack of logistic and proper coordination, (2) overproduction of IV antimicrobials due to poor communication regarding discharge dates and shortened duration of treatment, (3) delayed removal of peripherally inserted central catheters (PICC), increasing infection and thrombosis risk [4,14,15], and (4) insufficient antimicrobial stewardship, exemplified by overuse of reserve antibiotics, such as meropenem, and the inappropriate use of IV therapy where oral treatment was appropriate [11,16,17]. As is well-established, antimicrobial stewardship is essential in reducing antimicrobial resistance [18].

The structured OPAT programme introduced two key changes: (1) mandatory assessment of all OPAT requests by an ID specialist, providing personalized advice on antimicrobial therapy, with an emphasis on oral and narrow spectrum therapy when suitable, and monitoring, including therapeutic drug monitoring (TDM); and (2) an integrated workflow in the electronic health record (EHR), capturing all OPAT-related steps and communications across disciplines, and a well-organized logistical structure with a clear coordinating role for pharmacy and aftercare services. These conditions enable rapid delivery, smooth collaboration, and optimal continuity of care, all of which contribute to limiting unnecessary hospital stay [3,4].

This study evaluates the OPAT programme and its impact on duration of IV antimicrobial therapy, clinical response, recurrences, complications, readmissions, timely PICC removal, and healthcare costs before (2019) and after (2022–2024) implementation, as well as the effect of mandatory ID specialist assessment on antimicrobial stewardship parameters.

## 2. Results

### 2.1. Patient Characteristics

In total, 529 patients were included in the study, after excluding 9 patients because of missing data due to transfer to another hospital. There were 118 (22.3%) patients who had received treatment during the pre-OPAT period and 411 (77.7%) patients during the OPAT period. Table 1 provides an overview of the patient characteristics and patient outcomes for both treatment groups.

The distribution of hospital departments where the patients were admitted differed significantly between both treatment groups (*p* = 0.002). Patients in the OPAT group were more frequently admitted to the department of cardiology (17.5% vs. 8.5%), geriatrics (6.8% vs. 1.7%), and orthopedics (5.1% vs. 0.8%), while the pre-OPAT group contained more patients who were admitted to the department of urology (20.3% vs. 8.5%).

Additionally, significant differences were observed in the primary diagnoses between the groups (*p* = 0.001). Endocarditis (22.9% vs. 11.9%) and bloodstream infections (13.9% vs. 5.9%) were more common in the OPAT group, whereas urinary tract infections were more frequent in the pre-OPAT group (29.7% vs. 14.4%).

A higher proportion of patients with Staphylococcus aureus bacteraemia was found in the OPAT group (25.5% vs. 12.7%; *p* = 0.005), whereas MDROs were more prevalently detected in the pre-OPAT group (18.6% vs. 10.2%; *p* = 0.021). However, no significant difference between the groups were found in other bacteraemia (*p* = 0.925) or infections involving prosthetic material (*p* = 0.538).

### 2.2. Patient Outcomes

Table 2 shows the crude and adjusted model of the associations between the outcome variables and the implementation of OPAT. The crude model shows the results of the univariable analyses, whereas the adjusted model shows the results of the multivariable analyses in which the associations are adjusted for potential confounders.

As shown in Table 1, patients in the OPAT group had a shorter duration of IV therapy with a mean of 21.8 days (SD 18.1) compared to 34.4 days (SD 38.5) in the pre-OPAT group (*p* = 0.001). Notably, the larger variation in treatment duration observed in the pre-OPAT group (SD 38.5 vs. 18.1) suggests that OPAT implementation contributed to a more predictable and standardized course of therapy. The reduction in IV therapy duration was supported by the adjusted regression analysis, with a corresponding adjusted regression coefficient of −13.97 (95%CI −18.79 to −9.15; *p* < 0.001), as shown in Table 2.

A good clinical response was reported in 66.9% of OPAT patients and in 68.6% of pre-OPAT patients, whereas poor outcomes occurred in 9.7% (OPAT) and 11.9% (pre-OPAT), and these differences were not statistically significant (*p* = 0.313). This was confirmed by the adjusted model showing no statistically significant difference as well (adjusted OR: 1.22; 95%CI: 0.67–2.32; *p* = 0.527).

Additionally, recurrences occurred less frequently in the OPAT group (9.0% vs. 17.8%; *p* = 0.011), although this association was no longer statistically significant after adjustment (adjusted OR: 0.74; 95%CI 0.38–1.48; *p* = 0.353). Complications were also significantly less common in the OPAT group. Mild complications were reported in 8.5% and severe complications in 2.9% of the OPAT patients, compared to 12.7% and 8.5%, respectively, in the pre-OPAT group (*p* = 0.009). The multivariable analysis confirmed this association with an adjusted OR of 0.44 (95%CI 0.24–0.81; *p* = 0.007).

No significant difference was observed in readmission rates (18.5% vs. 24.6%), with a corresponding adjusted OR of 0.75 (95%CI 0.44–1.29; *p* = 0.287). Additionally, the timely removal of the PICC was reported more often in the OPAT group (93.3%) compared to the pre-OPAT group (80.6%; *p* = 0.002), among the 370 patients with available data on this variable. The adjusted model confirmed this association with an adjusted OR of 0.29 (95%CI 0.12–0.68; *p* = 0.004).

Furthermore, the combined variable of adverse clinical outcomes showed that OPAT patients had significantly lower odds of experiencing such adverse outcomes compared to patients in the pre-OPAT group, with an adjusted OR of 0.58 (95%CI 0.37–0.92; *p* = 0.021).

### 2.3. The Impact of the ID Specialist Assessment

ID specialist assessment of OPAT requests resulted in adjustment in therapy in 149 (36.3%) OPAT patients. In 27.8% of cases (n = 114), an oral alternative was available, of which 25.1% (n = 103) were actually prescribed. In the pre-OPAT cohort, retrospective assessment showed that an oral alternative would have been available in 33.1% (n = 39) of cases if an OPAT team had been in place at the time. Additionally, ID specialist assessment in OPAT patients led to discontinuation of antibiotics in 2.7% (n = 11), adjustment in dosage or duration in 4.1% (n = 17), and an alternative (narrower spectrum) IV antibiotic in 4.4% (n = 18). In pre-OPAT patients, this would have been possible in 0.8% (n = 1), 16.9% (n = 20), and 5.9% (n = 7), respectively. Furthermore, the assessment estimated that antibiotic treatment duration could have been shortened in 52 (44%) pre-OPAT patients by a median of 16 days (IQR: 12.0–28.0).

Appendix A presents the annual number of days each IV antimicrobial was used per year. A significant reduction in the use of the reserve antibiotic meropenem was observed, from 801 treatment days (26.5%) in the pre-OPAT cohort to 84 (11.9%), 175 (11.8%), and 121 (6.0%) treatment days in the OPAT cohort in 2022, 2023, and 2024, respectively, (*p* < 0.001). Mean treatment duration of meropenem was significantly shorter in the OPAT group, 14.9 (SD 14.0) days vs. 35.1 (SD 46.1) days pre-OPAT (*p* = 0.032), while small-spectrum agents such as benzylpenicillin and amoxicillin were used more frequently in the OPAT cohort (*p* < 0.001). Furthermore, as shown in Figure 1, due to IV to oral switch, low-dose flucloxacillin use declined from 298 treatment days (9.9%) in the pre-OPAT cohort to 55 (7.8%), 88 (5.9%), and 82 (4.1%) treatment days in the OPAT cohort in 2022, 2023, and 2024, respectively (*p* < 0.001). Cefuroxime use was also reduced in the OPAT cohort (*p* < 0.001).

Among OPAT patients, PICC placement was clinically indicated for IV antimicrobials in 68.1% (n = 280) and was avoided following IV to oral switch in 18.2% (n = 75), 5.8% (n = 24) of OPAT patients needed the PICC for other indications, and 7.8% (n = 32) were placed without a clear clinical indication, usually prior to ID specialist consultation. The number of OPAT consults required to prevent one PICC placement was 5.5 (95%CI 4.5–6.9), which could be decreased to 3.8 (95%CI 3.3–4.6) when no PICCs were placed before ID specialist consultation.

### 2.4. Costs

Table 3 provides an overview of the costs and savings associated with the implementation of the OPAT programme. Cost-reducing factors included reduced IV preparations, increased use of oral antimicrobials, TDM procedures, and PICC placements and a shorter duration of discharge procedure. In contrast, prescriptions of ertapenem and the costs of the OPAT team contributed to higher expenses. Overall, the implementation of the OPAT programme resulted in reduced costs of 3.343 EUR per patient compared to the pre-OPAT period.

## 3. Discussion

Instituting antimicrobial stewardship measures is essential in combating emerging antimicrobial resistance [18,20,21]. We found that implementation of a structured OPAT programme, including mandatory ID specialist assessment, was associated with improvements in several key antimicrobial stewardship parameters. In the OPAT cohort, therapy was adjusted in 36.3% of patients, and the duration of IV antimicrobial therapy was significantly shorter by 14 (mean) days, whereas no significant difference was found in clinical response. Adverse clinical outcomes were significantly less common in the OPAT cohort. In addition, the OPAT programme led to cost savings of 3.343 EUR per patient.

To our knowledge, this is the first study evaluating patient outcomes together with healthcare costs and the contribution of ID specialist assessment on antimicrobial stewardship aspects after implementing a structured OPAT programme, comparing this to prior outpatient care, while most studies compare outpatient to inpatient antimicrobial therapy [4,7,9,22], or compare IV to oral antimicrobials [23]. In the OPAT cohort, many patients received oral antibiotics, supporting evidence that oral therapy is equally effective as IV for several infections [16,17].

This study also provides valuable insights into the availability and application of (oral) alternatives for certain IV antibiotics. For example, the use of IV flucloxacillin has reduced from 456 days in the pre-OPAT year to 315 and 222 days, respectively, in 2023 and 2024, due to early IV to oral switch. Similarly, cefuroxime could almost completely be replaced by its oral form, reducing IV cefuroxime use from 211 days in the pre-OPAT year compared to just 55 days over 2.5 years during the OPAT period. Additionally, vancomycin was often replaced with oral alternatives such as fusidic acid or (prolonged) linezolid [24].

Another important example is the significant reduction in meropenem (*p* < 0.001), a last-resort antibiotic, from 801 treatment days in the pre-OPAT year to 175 in 2023 and 121 in 2024. These reductions resulted from ID specialist assessment, recommending alternative oral antibiotics instead of meropenem, such as Fosfomycin [25], pivmecillinam, or cefpodoxime + (amoxicillin)/clavulanic acid. Unfortunately, meropenem is globally prescribed more often, which is concerning considering the increase in antimicrobial resistance [21]. Using appropriate alternatives, as demonstrated in this study, is crucial, not only to limit resistance but also to reduce costs and improve patient satisfaction, which aligns with the principles of antimicrobial stewardship [11,26,27,28]. Integration of mandatory ID specialist assessment within the OPAT programme ensures optimal antibiotic use [23], which is consistent with the study of Wijnaker and colleagues, who reported that in 28% of cases ID specialist consultation would have led to adjustment in treatment [29], and Cassettari and colleagues, who reported an oral alternative in 24.7% of cases [30].

Furthermore, the OPAT programme resulted in substantial cost savings of 3.343 EUR per patient. The IV to oral switch contributed significantly to overall cost reduction, since costs for IV antibiotics are substantially higher. Several studies have reported reduced healthcare costs associated with OPAT [2,4,23], which is consistent with our study. Although not directly quantifiable in financial terms in our study, OPAT generates additional economic benefits for healthcare by shifting care from the hospital to the home setting [2,4,23]. This underscores the importance of effective collaboration among all involved healthcare providers.

An area for improvement includes reducing unnecessary PICC placements. As demonstrated, there were still 32 (7.8%) unnecessary PICC placements in the OPAT cohort, which often occurred before OPAT team consultation, after which the ID specialist assessment recommended oral therapy instead. This highlights the importance of consulting the OPAT team prior to PICC placement.

Future developments could include outpatient PICC placement, eliminating the need for hospital admission, and self-administered OPAT (S-OPAT) to reduce healthcare burden, costs, and increase patient autonomy [31]. Several hospitals are implementing S-OPAT and are currently researching its safety and effectiveness. While some studies report increased adverse events, the overall findings are promising [31].

Limitations include the analyses spanning different time periods, during which variations in the population may have introduced bias. Although the retrospective evaluation of antibiotic alternatives and adjustments for the pre-OPAT cohort provides valuable insights, retrospective assessments have limitations such as bias and could only be based on information documented in EHR during admission. 

Also, our study was a single-centred study in a regional hospital in The Netherlands, limiting the generalizability of the findings to healthcare facilities in countries with other healthcare systems and patient populations. However, within The Netherlands the SG can be seen as a representative large hospital.

Furthermore, missing data on the timely PICC line removal, usually carried out by outpatient care providers, could have underestimated the risk of PICC infections and complications.

Strengths include the comprehensive evaluation covering quality of care, clinical outcomes and cost evaluation, and adjustment for confounders. Furthermore, the study included all patients who received outpatient parenteral antimicrobial therapy and received follow-up in SG. No selection criteria were applied, resulting in a study population with a wide variety of patient characteristics and clinical indications. 

## 4. Materials and Methods

### 4.1. Study Design

We conducted a retrospective before–after cohort study comparing patients treated according to the newly introduced OPAT programme (August 2022–December 2024) with a pre-implementation cohort (pre-OPAT) from 2019. The pre-OPAT year was chosen to avoid the confounding effects of the COVID-19 pandemic. The study was conducted at SG, a large regional teaching hospital with more than 600 beds in Haarlem and Hoofddorp, The Netherlands. The primary objective was to evaluate differences in patient outcomes and healthcare costs between the two cohorts. The study was approved by the Institutional Review Board of SG (ACLU 2024.0136) and exempt from requiring informed consent.

### 4.2. OPAT Programme Procedure

The primary physician requests OPAT in EPIC, the EHR used in SG, for patients receiving IV or intraperitoneal antibiotics and who are eligible for outpatient treatment, after which ID specialist assessment is performed, providing advice on antimicrobial choice, dosage, administration route, duration, date of PICC removal, and laboratory tests and TDM. If IV therapy was indicated and approved, a PICC was placed, discharge was arranged, and outpatient care coordinated. The outpatient pharmacy supplies oral and/or IV antimicrobials, guides patients, monitors medication, and advises on dosing.

### 4.3. Study Population

Patients were included in the pre-OPAT cohort if an outpatient care order for IV or intraperitoneal antimicrobial treatment had been placed for them in 2019. Both patients who received outpatient therapy and those who remained hospitalized after the request were included. The OPAT cohort included patients for whom an OPAT order was requested between the start of the programme on 15 August 2022 and 31 December 2024. This included both patients who received OPAT as well as patients for whom OPAT was not indicated after ID specialist assessment was effectuated, because antimicrobial treatment was completed during hospitalization, an oral alternative was available, or antimicrobial treatment was unnecessary. The study population included patients of all ages admitted to different departments with various infectious diseases. Patients with missing outcome data due to transfer without follow-up were excluded.

### 4.4. Data Collection

Patient data were extracted from EPIC and GLIMS, the laboratory information system of RPHLK. Prior to the data collection, variables were selected to reflect the potential impact of the OPAT programme, based on clinical relevance and literature review.

### 4.5. Variables

Pre-OPAT vs. OPAT treatment periods were compared. Outcomes identified to analyze the potential impact of the OPAT programme included duration of IV antibiotic therapy, clinical response, recurrences, complications, readmissions, timely PICC removal, and composite variable adverse clinical outcomes, comprising variables such as recurrences, complications, and readmissions.

Definitions:Clinical response: Assessed by the study team based on documentation of treating physicians when antimicrobial therapy was completed, categorized into four groups: good (no residual symptoms/negative culture), reasonable (minor symptoms requiring no additional treatment), poor (severe symptoms/readmission), or deceased.Recurrence: Reinfection with the same microorganism within 3 months of therapy completion.Complications: Included both adverse drug reactions and catheter-associated complications (infection and thrombosis). Categorized as mild, severe (hospitalization, discontinuation of therapy, or death) or none.Readmission: Unplanned admission during or ≤3 months after completing antimicrobial therapy.Timely PICC removal: Within 3 days of IV completion, or not placed, or needed for other medical treatments.

Potential confounders included the following: sex (may influence treatment duration), age, department, infection type/severity, presence of foreign material, bacteraemia, or MDROs.

Primary indication was defined as the main reason for antibiotic therapy. In case of multiple indications, the indication associated with the longest IV treatment duration was chosen as the primary indication. Additional indications were defined as factors that added to the complexity of the primary infection, including bacteraemia, the presence of foreign material, or infection with MDROs. Patients could have none, one, or multiple additional indications.

### 4.6. Evaluation of ID Specialist Assessment

We documented all therapy adjustments resulting from ID assessment in the OPAT period (switch to an oral alternative, discontinuation of antibiotic therapy, alternative IV antibiotic (narrower spectrum), or adjustments in dosage or treatment duration). For the pre-OPAT cohort, two ID specialists retrospectively assessed whether adjustments would have been made if the OPAT programme had been in place at the time, based on knowledge, guidelines, and availability of antibiotics applicable for that period. This assessment was conducted to ensure that the number of alternatives and adjustments in the OPAT cohort could not only be attributed to improved guidelines or increased availability of treatment options over the years. Furthermore, PICC placements were categorized as avoided, clinically appropriate, or unnecessarily inserted.

### 4.7. Cost Analysis

Cost data were obtained from hospital administrative sources. The internal medicine department’s average daily admission cost was used as the inpatient cost benchmark. Costs associated with outpatient care during pre-OPAT and OPAT periods were compared, including costs of preparation of IV antibiotics, oral antibiotics, PICC placements, TDM, OPAT team, and the number of days waiting for discharge (days from outpatient care order to discharge). Calculated costs for IV and oral antibiotics are based on per-day prices and treatment duration (Appendix A). Costs were adjusted for inflation and converted to 2024 using inflation rates to correct for price changes over time [19]. For both groups, costs were calculated per patient and for cohorts of 100 patients. The difference between both periods was interpreted as the cost savings. Only direct hospital and pharmacy costs were included.

### 4.8. Statistical Analyses

Baseline characteristics and outcome variables were compared between the cohorts using Chi-square or Fisher’s exact tests for categorical variables and independent *t*-tests or Mann–Whitney U-tests for continuous variables. Categorical variables were reported as frequencies (n) and percentages (%) and continuous variables as means with standard deviations (SD) or medians with interquartile ranges (IQR).

To analyze the associations between the outcome variables and the independent variable, separate univariable analyses were performed for each outcome variable: logistic regression was used for the categorical outcomes and linear regression for the continuous outcome. Prior to the analyses, the variables clinical response and complications were dichotomized by combining good and reasonable response, and severe and mild complications. Results were reported as odds ratios (OR) with 95% confidence intervals (CI) and corresponding *p*-values for logistic regressions, or as regression coefficients (B) for the linear regression. Subsequently, a multivariable analysis was performed for each association to adjust for the potential confounders. The Cochran–Armitage test for trend was used to assess whether there was a significant linear trend in the use of IV antimicrobials over the years. Analyses were performed with R version 4.0.3. Significance was set at *p* < 0.05.

There was no funding source for this study.

## 5. Conclusions

Overall, the findings of this study indicate that a structured OPAT programme, including mandatory ID specialist assessment, has a significant positive impact on patient outcomes, reducing healthcare costs and optimizing antimicrobial use, and can therefore be an important tool for antimicrobial stewardship. Furthermore, this study identifies some areas of improvement and addresses ideas for future implementation of OPAT.

## Figures and Tables

**Figure 1 antibiotics-14-01103-f001:**
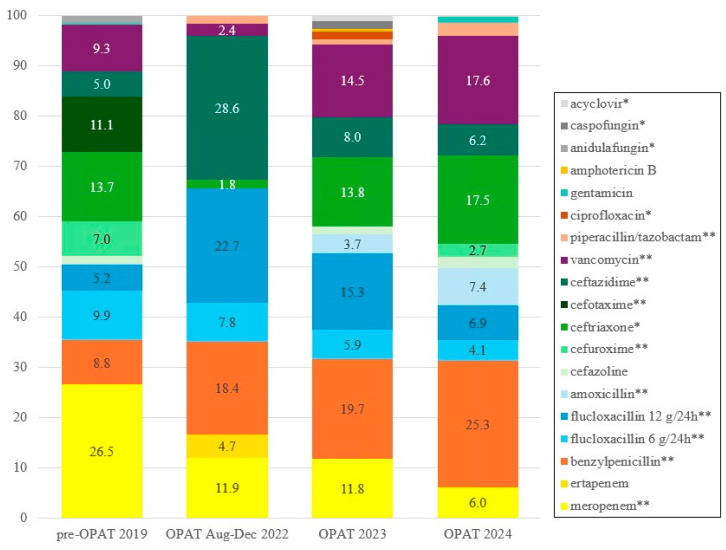
IV antimicrobial use per year (%). * = *p* < 0.05; ** = *p* < 0.001 using Cochran–Armitage test for trend. A complete overview of IV antimicrobial use per year is found in Appendix A.

**Table 1 antibiotics-14-01103-t001:** Patient characteristics and patient outcomes, stratified by cohort (pre-OPAT vs. OPAT).

	Total n = 529	Pre-OPAT n = 118	OPAT n = 411	*p*-Value
**Sex** n (%)				0.360
Male	335 (63.3%)	70 (59.3%)	265 (64.5%)	
Female	194 (36.7%)	48 (40.7%)	146 (35.5%)	
**Age** mean (SD)	67.3 (17.5)	65.3 (16.5)	67.8 (17.7)	0.155
**Department** n (%)				**0** **.** **002**
Internal medicine (incl. Nephrology, Oncology)	152 (28.7%)	39 (33.1%)	113 (27.5%)	
Surgery (incl. Vascular Surgery and Trauma Surgery)	76 (14.4%)	18 (15.3%)	58 (14.1%)	
Cardiology	82 (15.5%)	10 (8.5%)	72 (17.5%)	
Pulmonology	41 (7.8%)	11 (9.3%)	30 (7.3%)	
Orthopedics	22 (4.2%)	1 (0.8%)	21 (5.1%)	
Urology	59 (11.2%)	24 (20.3%)	35 (8.5%)	
Geriatrics	30 (5.7%)	2 (1.7%)	28 (6.8%)	
Neurology (incl. Neurosurgery)	38 (7.2%)	7 (5.9%)	31 (7.5%)	
Pediatrics	13 (2.5%)	3 (2.5%)	10 (2.4%)	
Other departments ^1^	16 (3.0%)	3 (2.5%)	13 (3.2%)	
**Primary indication** n (%)				**0** **.** **001**
Endocarditis	108 (20.4%)	14 (11.9%)	94 (22.9%)	
Urinary tract infection	94 (17.8%)	35 (29.7%)	59 (14.4%)	
Pulmonary infection	43 (8.1%)	12 (10.2%)	31 (7.5%)	
Abscess/soft tissue infection	30 (5.7%)	3 (2.5%)	27 (6.6%)	
Neurological infection/eye infection	39 (7.4%)	11 (9.3%)	28 (6.8%)	
Osteomyelitis/spondylodiscitis/arthritis	98 (18.5%)	25 (21.2%)	73 (17.8%)	
Vascular infection/aortitis/mycotic aneurysm	26 (4.9%)	5 (4.2%)	21 (5.1%)	
Bloodstream infection (without focus, or associated with phlebitis or intravascular catheter-related origin)	64 (12.1%)	7 (5.9%)	57 (13.9%)	
Other infections ^2^	27 (5.1%)	6 (5.1%)	21 (5.1%)	
**Additional indication** n (%)				
*S. aureus* bacteraemia	120 (22.7%)	15 (12.7%)	105 (25.5%)	**0** **.** **005**
Other bacteraemia or candidemia	152 (28.7%)	33 (28.0%)	119 (29.0%)	0.925
Device related infections ^3^	107 (20.2%)	21 (17.8%)	86 (20.9%)	0.538
Multidrug-resistant organisms	64 (12.1%)	22 (18.6%)	42 (10.2%)	**0** **.** **021**
**Duration of IV antibiotics** (days) mean (SD)	24.7 (24.7)	34.4 (38.5)	21.8 (18.1)	**0** **.** **001**
**Clinical response** n (%)				0.313
Good	356 (67.3%)	81 (68.6%)	275 (66.9%)	
Reasonable (still some symptoms)	87 (16.4%)	20 (16.9%)	67 (16.3%)	
Poor	54 (10.2%)	14 (11.9%)	40 (9.7%)	
Deceased ^4^	32 (6.0%)	3 (2.5%)	29 (7.1%)	
**Recurrence (<3 months)** n (%)				**0** **.** **011**
No	471 (89.0%)	97 (82.2%)	374 (91.0%)	
Yes	58 (11.0%)	21 (17.8%)	37 (9.0%)	
**Complications** n (%)				**0** **.** **009**
No complications	457 (86.4%)	93 (78.8%)	364 (88.6%)	
Mild complications	50 (9.5%)	15 (12.7%)	35 (8.5%)	
Severe complications	22 (4.2%)	10 (8.5%)	12 (2.9%)	
**Readmissions (<3 months)** n (%)				0.184
No	424 (80.2%)	89 (75.4%)	335 (81.5%)	
Yes	105 (19.8%)	29 (24.6%)	76 (18.5%)	
**Timely removal of PICC** n (%) *				**0** **.** **002**
Yes	336 (90.8%)	58 (80.6%)	278 (93.3%)	
No	34 (9.2%)	14 (19.4%)	20 (6.7%)	
**Days waiting for discharge** mean (SD) **	4.8 (6.0)	4.7 (6.3)	4.8 (5.8)	0.836

Statistically significant *p*-values are presented in bold. * Only patients with available data on PICC removal were included (n = 370; 69.9%). ** Only patients who received outpatient antibiotic therapy were included (n = 459; 86.8%). ^1^ Intensive care, plastic surgery, rheumatology, gastroenterology, or general practice. ^2^ e.g., pancreatitis, cholangitis, peritonitis, or Lemierre syndrome. ^3^ Infection of prosthesis, osteosynthesis material, artificial devices, or cardiac devices. ^4^ Due to an underlying medical condition, unrelated to outpatient intravenous antibiotic therapy.

**Table 2 antibiotics-14-01103-t002:** Crude and adjusted model of the association between the independent variable (pre-OPAT vs. OPAT) and the outcome variables.

	Baseline	Model			Adjusted	Model *		
	Regression Coefficient	Odds Ratio	CI 95%	*p*-Value	Regression Coefficient	Odds Ratio	CI 95%	*p*-Value
Duration of IV therapy	−12.58		[−17.55; −7.62]	**<0** **.** **001**	−13.97		[−18.79; −9.15]	**<0** **.** **001**
Clinical response		1.20	[0.69–2.19]	0.537		1.22	[0.67–2.32]	0.527
Recurrences		0.46	[0.26–0.83]	**0** **.** **008**		0.74	[0.38–1.48]	0.353
Readmissions		0.70	[0.43–1.15]	0.145		0.75	[0.44–1.29]	0.287
Complications		0.48	[0.28–0.83]	**0** **.** **007**		0.44	[0.24–0.81]	**0** **.** **007**
Timely PICC removal **		0.30	[0.14–0.63]	**0** **.** **001**		0.29	[0.12–0.68]	**0** **.** **004**
Combination variable ***		0.48	[0.31–0.73]	**0** **.** **001**		0.58	[0.37–0.92]	**0** **.** **021**

Statistically significant *p*-values are presented in bold. The baseline model presents the univariable regression analyses. The adjusted model presents the multivariate regression analyses, adjusted for potential confounders. * Adjusted for department, primary indication, extra indication, age, and sex. ** Only patients with available data on PICC removal were included (n = 370; 69.9%). *** Reflecting the presence of one or more recurrences, readmissions, and complications.

**Table 3 antibiotics-14-01103-t003:** Overview of costs and savings in the pre-OPAT and OPAT periods presented per patient and per 100 patients, corrected for inflation [19].

	Pre-OPAT		OPAT		Difference	
	*Per patient*	*Per 100 patients*	*Per patient*	*Per 100 patients*	*Per patient*	*Per 100 patients*
IV preparations	5,201.18	520,117.76	1,879.38	187,938.08	3,321.80	332,179.68
Oral antibiotics			24.14	2,414.00	- 24.14	- 2,414.00
TDM ^1^	1.24	124.40	1.16	115.70	0.08	8.70
Ertapenem *			15.59	1,559.12	- 15.59	- 1,559.12
OPAT team			23.43	2,343.00	- 23.43	- 2,343.00
PICC placements	234.53	23,453.39	186.13	18,613.13	48.40	4,840.26
Days waiting for discharge	2,928.67	292,867.44	2,892.46	289,246.02	36.21	3,621.42
**Total**	8,365.62	836,562.99	5,022.29	502,229.05	3,343.33	334,333.94

All amounts are reported in euros (EUR). * Ertapenem is at the expense of the hospital department, unlike the other prescribed antibiotics, which are reimbursed by the health insurance provider. It is only prescribed in one patient during the study period. ^1^ TDM = therapeutic drug monitoring.

## Data Availability

When requested, deidentified data will be made available with publication, and after approval of a proposal, with a signed data access agreement.

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
