# Peer review of "The Impact of a Structured Outpatient Parenteral Antimicrobial Therapy (OPAT) Programme on Quality of Care, Optimisation of Antimicrobial Use, and Healthcare Costs: A Retrospective Cohort Study"

_antibiotics, 2025, doi:10.3390/antibiotics14111103_

Round 1

Reviewer 1 Report

Comments and Suggestions for Authors

The authors demonstrate measurable benefits in antimicrobial stewardship, patient safety, and cost reduction. The manuscript is well written, clear in its objectives, and supported by appropriate analyses. With the following revisions, it would make a strong contribution to the OPAT stewardship literature.

  1. In Table 1, variables with statistically significant p-values are not visually distinguished. I recommend highlighting significant rows (e.g., by bolding values, adding asterisks, or including an explanatory footnote such as *p < 0.05; **p < 0.01; *** p < 0.001).
  2. Figure 1 lacks indicators of variability (e.g., error bars) or statistical significance and a visual takeaway. Too many antibiotics are plotted together, and the color scheme and labels are difficult to interpret. The legend also does not specify what the percentages represent or which statistical method was applied. I recommend replacing the current chart with a simplified grouped or stacked bar chart
  3. The classification of clinical response as good, reasonable, or poor appears subjective and may vary depending on the clinician documentation. Please clarify whether these assessments were blinded to the study period and who determined them (e.g., treating physician, OPAT team).
  4. The title is strong and informative, but overly conclusive in tone for a retrospective before–and–after study. Rephrasing it to emphasize evaluation or impact assessment rather than proven improvement would make it more scientifically precise and suitable.

Author Response

Thank you for reviewing our work and for the helpfull feedback. 

Comments 1: In Table 1, variables with statistically significant p-values are not visually distinguished. I recommend highlighting significant rows (e.g., by bolding values, adding asterisks, or including an explanatory footnote such as *p < 0.05; **p < 0.01; *** p < 0.001).

Response 1: Statistically significant p-values were made bold in the tables. 

Comments 2: Figure 1 lacks indicators of variability (e.g., error bars) or statistical significance and a visual takeaway. Too many antibiotics are plotted together, and the color scheme and labels are difficult to interpret. The legend also does not specify what the percentages represent or which statistical method was applied. I recommend replacing the current chart with a simplified grouped or stacked bar chart

Response 2: Unfortunately, the figures and tables we submitted appear to have been edited in a way that rendered the legends unreadable. Stacked bar chart is made, and asterixes were placed for statistically significant change in antibiotic use, using Cochran-Armitage test for trend. See page 7, line 187. 

Figure 1: IV antimicrobial use per year (%).
* = p<0.05; ** = p<0.001 using Cochran-Armitage test for trend.
A complete overview of IV antimicrobial use per year is found in Table A4.

Comments 3: The classification of clinical response as good, reasonable, or poor appears subjective and may vary depending on the clinician documentation. Please clarify whether these assessments were blinded to the study period and who determined them (e.g., treating physician, OPAT team).

Response 3: Thank you for pointing this out. To clarify we adjusted the sentence in line 324-328 on page 10. 

Clinical response: assessed by the study team based on documentation of treating physicians when antimicrobial therapy was completed, categorized into four groups: good (no residual symptoms/negative culture), reasonable (minor symptoms requiring no additional treatment), poor (severe symptoms/readmission), or deceased. 

Comments 4: The title is strong and informative, but overly conclusive in tone for a retrospective before–and–after study. Rephrasing it to emphasize evaluation or impact assessment rather than proven improvement would make it more scientifically precise and suitable.

Response 4: We rephrased the title, see page 1, line 2. 

The impact of a structured Outpatient Parenteral Antimicrobial Therapy (OPAT) programme on quality of care, optimisation of antimicrobial use, and healthcare costs: a retrospective cohort study.

Reviewer 2 Report

Comments and Suggestions for Authors

Thank you to the editors for entrusting me with the manuscript for review.

The work has been prepared correctly and is valuable material in the context of optimizing pharmacotherapy with antibiotics. The results of the study indicate the possibility of improvement in this area, which is an extremely important issue.

It is worth noting that the authors reliably point out not only the achievements of the work, but also its limitations. The section describing the OPAT procedure is well prepared and may be a valuable source of information for teams interested in implementing similar solutions in their facilitie.

However, I have doubts about the availability of specialists with sufficient clinical experience who could more commonly perform the role of OPAT specialists. The limited number of such specialists could lead to delays in the implementation of sometimes necessary antibiotic therapy. If the authors have knowledge in this area, it is worth referring to it. If the authors do not feel competent in this matter, we will leave this question open.

I also have a few minor editorial suggestions that may make it easier for readers to understand the paper:

The introduction (line 23) contains the abbreviation PICC, which is not explained until line 68. I suggest explaining the abbreviation in the introduction, as this is often the only part of the paper that is analyzed during a quick review of the literature before reading the entire text.

In Tables 1 and 2, the dots in the numerical values are in the middle instead of at the bottom. Can this be changed?

Fig. 1 explains the meaning of only a few colors. Can you expand the description to include the other colors used in the graph?

In line 127, where the duration of pre-OPAT and OPAT therapy was analyzed, I think it is also worth noting the greater variation in the duration (SD) of antibiotic therapy in the pre-OPAT group. This is an important aspect because it shows improved predictability in the course of treatment.

I also suggest arranging Abbreviations alphabetically.

I recommend publishing the work after minor corrections have been made.

Author Response

Thank you for reviewing our manuscript and providing valuable feedback.

Comments 1: However, I have doubts about the availability of specialists with sufficient clinical experience who could more commonly perform the role of OPAT specialists. The limited number of such specialists could lead to delays in the implementation of sometimes necessary antibiotic therapy. If the authors have knowledge in this area, it is worth referring to it. If the authors do not feel competent in this matter, we will leave this question open.

Response 1:  Thank you for pointing this out. We added a sentence showing that 78% of OPAT teams have ID specialist involvement. See line 59, page 2, paragraph 2 of the revised manuscript: 

Nevertheless, structured implementation is limited [11]; only a minority of hospitals in the world have dedicated OPAT teams (in 42%) [12], selection criteria (in 56.4%), and follow-up protocols (61.8%) [13]. Among hospitals with OPAT teams, 78% have an infectious disease (ID) specialist involved [12].   

Comments 2: The introduction (line 23) contains the abbreviation PICC, which is not explained until line 68. I suggest explaining the abbreviation in the introduction, as this is often the only part of the paper that is analyzed during a quick review of the literature before reading the entire text.

Response 2: Agreed, we have changed this accordingly. See line 21-24, page 1: 

Our objective was to evaluate the impact of the OPAT programme on patient outcomes (IV duration, clinical response, adverse clinical outcomes, timely peripherally inserted central catheter (PICC) removal), antimicrobial stewardship parameters, and healthcare costs. 

Comments 3: In Tables 1 and 2, the dots in the numerical values are in the middle instead of at the bottom. Can this be changed?

Response 3: Changed to dots at the bottom. 

Comments 4: Fig. 1 explains the meaning of only a few colors. Can you expand the description to include the other colors used in the graph?

Response 4: Unfortunately, the figures and tables we submitted appear to have been edited in a way that rendered the legends unreadable. We will provide the tables and figures as separate files to ensure proper display. 

Comments 5: In line 127, where the duration of pre-OPAT and OPAT therapy was analyzed, I think it is also worth noting the greater variation in the duration (SD) of antibiotic therapy in the pre-OPAT group. This is an important aspect because it shows improved predictability in the course of treatment.

Response 5: Thank you for this suggestion. We revised this part as shown below in line 129-131, page 5, paragraph 1. 

As shown in Table 1, patients in the OPAT group had a shorter duration of IV therapy with a mean of 21.8 days (SD 18.1) compared to 34.4 days (SD 38.5) in the pre-OPAT group (p=0.001). Notably, the larger variation in treatment duration observed in the pre-OPAT group (SD 38.5 vs. 18.1) suggests that OPAT implementation contributed to a more predictable and standardized course of therapy. The reduction in IV therapy duration was supported by the adjusted regression analysis, with a corresponding adjusted regression coefficient of –13.97 (95%CI –18.79 to –9.15; p<0.001), as shown in Table 2.

Comments 6:  I also suggest arranging Abbreviations alphabetically.

Response 6: Changed accordingly. See page 12, line 406.